# Development and Consumer Perception of a Snack Machine Producing Customized Spoonable and Drinkable Products Enriched in Dietary Fiber and Protein

**DOI:** 10.3390/foods9101454

**Published:** 2020-10-13

**Authors:** Kaisa Vehmas, Alex Calton, Katri Grenman, Heikki Aisala, Nesli Sozer, Emilia Nordlund

**Affiliations:** VTT Technical Research Centre of Finland Ltd., P.O. Box 1000, FI-02044 VTT, Finland; alex.calton@vtt.fi (A.C.); katri.grenman@vtt.fi (K.G.); heikki.aisala@vtt.fi (H.A.); nesli.sozer@vtt.fi (N.S.); emilia.nordlund@vtt.fi (E.N.)

**Keywords:** healthy snacking, prototype, personalized food, customization, consistency, co-creation, user experience, consumer liking

## Abstract

The aim of the study was to evaluate consumer perceptions toward customized snacks produced with a Healthy Snack Machine (HSM) prototype, at-site of the purchase and consumption. The present study had a multi-disciplinary approach including both snack product and HSM development (hardware and user interface). Snack development included both instrumental (viscosity, colloidal stability) and sensory characterization (by trained sensory (*N* = 10) and consumer (*N* = 55) panels) of spoonable and drinkable, oat- and dairy-based snack products, fortified with protein and/or dietary fiber. The protein and fiber addition reduced viscosity in spoonable products but did not affect the consistency of drinkable samples. Oat-based samples differed from dairy-based in multiple attributes in sensory profiling. In consumer sample testing, sample odor and taste were the most and least preferred aspects, respectively. In the snack machine testing, a qualitative consumer study (*N* = 33) showed that the HSM was easy to use, the user interface was clear, the ordering process was quick, and the participants were interested in using the HSM in the future. The snack choices (spoonable/drinkable and dairy/oat base) made by the consumers were distributed equally, but the berry-flavor was preferred over cocoa and vanilla. The most common HSM usage scenarios were “between work/school and hobbies” and “in transit from one place to another”.

## 1. Introduction

On-the-go eating and snacking are increasing, and it has been shown that snacking provides a large share of people’s daily energy intake. For example, in the United States, the average share of snacking is about 25% among children and 22% among adults [1]. Vending machines are common services providing snacks for on-the-go eating occasions [2]. However, the nutritional profile of the food products available in the vending machines is typically not optimal for supporting healthy diets. Instead, snack products are often pre-packed, energy-dense, and nutrient-poor [3].

Recent studies indicate strong consumer interest in healthier snacking [4,5], which, thus, provides an opportunity to develop new solutions to increase healthier food choices in the snacking category [6]. In our previous work, we investigated a healthy snacking and on-the-go eating concept, “Healthy Snack Machine” (HSM), that produces freshly made food and enables customization of the product at the site of purchase and consumption [7]. By using qualitative and quantitative research methods with an iterative consumer co-creation approach, we were able to show that consumers were attracted by a new snacking concept that would help them to consume healthier food and enable customization of the product. The time of the day and personal preferences affected HSM food choice and product customization, and consumers preferred satiating products that are convenient for on-the-go eating [7].

Personalization of on-the-go foods at the point of purchase for the individual consumer sets technological and process requirements for the food manufacturing. On-the-site formulation, e.g., in the HSM-type concept, emphasizes the role of ingredient properties that enable instant preparation of diverse personalized foods. Protein and dietary fiber are potential ingredients for the production of healthy on-the-go snacks, which can be included at elevated levels to provide subsequent health-supporting effects [8,9,10,11]. In our previous work, we evaluated the powder properties of ingredients that allow for instant production of variable textures with protein and dietary fiber supplementation levels, justifying a nutrition claim [12]. Required ingredient properties included free powder flow, specific particle size distribution, high porosity, and rapid de-agglomeration in water. Often, the ingredients do not have these required properties, which makes on-demand production of snacks challenging. In addition, on-demand production of ingredient mixes is not covered well in the scientific literature even though ingredients themselves are well studied. Our previous study concluded that from the ingredient behavior viewpoint, the snack consistency, as well as protein and dietary fiber content, could be customized for the targeted end products [12].

In the present study, the target was to evaluate consumers’ perceptions and liking toward customized snacks produced with the Healthy Snack Machine (HSM) prototype at-site of the purchase and consumption. In our previous study [7], the HSM concept was co-developed with consumers, and the study was implemented with mock-ups of the user interface (UI) of the HSM. The corresponding evaluation of virtual prototypes has been found as an efficient way of analyzing customer impressions and concretizes the service to the users before it is really available [7,13]. However, to proceed closer to real-life experimenting, in the present study, we developed a prototype of the HSM. Further, development of the HSM hardware was founded on the knowledge from Calton et al. [12] to enable instant production of spoonable and drinkable snack products that can be tailored according to protein and dietary fiber content.

Specifically, the aim of the present study was to (1) create the HSM prototype and UI for the snack machine, (2) investigate the rheological properties of the snack products produced by the HSM, (3) evaluate the textural and flavor attributes of the snacks by using a trained sensory and a consumer panel, as well as (4) study the overall user experience of the HSM by qualitative consumer testing of the prototype machine.

## 2. Materials and Methods

The present study had a multi-disciplinary approach including both snack product and HSM machine development (Figure 1). The snacks were developed and evaluated iteratively by a trained sensory panel and consumers during the food design process. The HSM machine concept included both hardware and user interface development. The developed HSM concept, including snack ordering by using the UI, machine usability, and snack properties, was evaluated by consumers.

### 2.1. Snack Product and Machine Prototype Development

#### 2.1.1. Snack Product Development and Characterization

Drinkable and spoonable snack products, either oat- or dairy-based, were prepared by mixing ingredients in powder form with water to enable instant manufacturing by the HSM at the point of consumption. The specific target for the snack development was to allow customization of the snack product to qualify for protein and/or dietary fiber nutrition claims [14], as illustrated for the cocoa-flavored samples in Table 1. Ingredients for snack preparation included cold water swelling (CWS) agglomerated waxy maize starch (NOVATION PRIMA^®^ 650) from Ingredion Germany GmbH (Hamburg, Germany); unflavored whey protein concentrate and lactose-free skim-milk powder from Valio Ltd. (Helsinki, Finland); digestion-resistant maltodextrin (Nutriose FM06) and pea protein isolates (Nutralys F85M and S85Plus D-EXP) from Roquette Frères (Lestrem, France); extruded oat endosperm flour and oat bran concentrate containing 15% β-glucan from Raisio plc (Raisio, Finland); dry malt extract from Muntons plc (Suffolk, UK); two cocoa powders (1) D-11-CK from ADM International Sàrl Cocoa Division (Rolle, Switzerland) and (2) 1856 from Condetta GmbH & Co KG (Halle, Germany); freeze-dried strawberry, wild bilberry, and lingonberry from Nature Lyotech Ltd. (Espoo, Finland); vanilla sugar from Mauste-Sallinen Ltd. (Naantali, Finland); chia seeds and vanilla-cranberry-strawberry granola from Risenta AB (Sollentuna, Sweden); and coconut flakes from Meira Ltd. (Helsinki, Finland).

The dairy- and oat-based base mixes were formulated as sweet and flavored with vanilla or cocoa (Table 2). Berry-flavored snacks consisted of the vanilla base combined with berry powder. Dairy- and oat-based cocoa-flavored snack samples were prepared using the HSM prototype for sensory profiling and consumer trials according to recipes presented in Table 2. After sensory profiling and consumer liking trials, the pea protein isolate for oat-based snack supplementation was replaced with a milder-flavored alternative (S85Plus D-EXP), and the base compositions were revised for the qualitative consumer testing of the HSM, as explained in Table 2.

Apparent viscosity and colloidal stability of the snack product samples were analyzed. The apparent viscosity of spoonable snacks prepared with the prototype machine were measured using an AR-G2 controlled stress rheometer (TA Instruments Ltd., Hertfordshire, UK) equipped with a Peltier standard cup and vane geometry (Vane Narrow gap, 998,080) as a function of the shear rate increasing from 0.1 to 200 s^−1^ at 15 °C. The duration of the measurement was 9 min with 10 measurement points per decade. Apparent viscosity values of duplicate measurements are presented at a shear rate of 31 s^−1^ at 30 and 60 min after preparation. Samples were stored at 7 °C before the measurement. The drinkable samples had low viscosity and, therefore, were not instrumentally determined. Results were presented as mean ± standard error and differences were analyzed with one-way ANOVA and Tukey’s HSD (honestly significant difference) post-hoc test using SPSS software, version 26 (IBM Corp, Armonk, NY, USA). Sedimentation tendency of drinkable samples was assessed as part of the descriptive sensory profiling of the snack products.

#### 2.1.2. Descriptive Sensory Profiling of the Snack Products

In the descriptive profiling of snacks, the focus was to study the effect of product type (spoonable and drinkable) and product base (dairy or oat) on the sensory profile of snacks with or without protein and fiber addition. The sensory properties of selected samples were studied with generic descriptive analysis by VTT’s trained food and beverage sensory panel (*N* = 10). The protocol for performing the sensory evaluation has been accepted by the Ethical Committee of VTT. All panelists gave their prior informed consent before the evaluations. The necessary individual assessor data were collected in accordance with the EU General Data Protection Regulation GDPR (2016/679). The samples included cocoa-flavored samples from both sample types (drinkable and spoonable) and the two base mixtures (dairy- and oat-based) either (1) with extra protein and dietary fiber or (2) without added protein or fiber. Cocoa flavoring was chosen as a representative flavor for the study. The samples were prepared by combining single portions produced by the HSM, dividing them into 30 mL samples, and refrigerating for up to one hour before evaluation. The base list of sensory attributes was formulated by four sensory experts. This list was further refined with all 10 assessors and the reference product intensities were tied to the attributes in a 1 h consensus session. The drinkable and spoonable samples were presented in the same panel training session but were evaluated in separate sessions. The attribute lists and reference products are shown in Appendix A (Appendix A). After panel training, the assessors evaluated the samples in two duplicate sessions using 0–10 line scales. The samples were presented in a complete block, Latin square design with three-digit codes. The data were collected using Compusense five version 5.6 (Compusense Inc., Guelph, ON, Canada). The data were analyzed with two-way mixed-model ANOVA using SPSS version 26 (IBM Corp., Armonk, NY, USA).

#### 2.1.3. Development of HSM Hardware and User Interface

The prototype machine hardware (Figure 2) of the HSM was based on a coffee machine targeted for workplaces, and it was rebuilt for the requirements of the HSM. The hardware consisted of solenoid valve-controlled water lines, auger-driven powder dispensers, and motorized mixing bowls (Figure 2a) controlled using a programmable logic controller implemented by an industrial embedded PC with the Windows operating system (Beckhoff Automation plc, Hyvinkää, Finland). Snack product preparation occurred in up to four steps (Figure 2b), where (1) a flavored base liquid was formed, (2) optional protein and/or dietary fiber was added to base, (3) optional thickener and/or berry powder was added, and (4) optional garnish was sprinkled into the serving cup. At the start of snack preparation, an opened solenoid valve allowed water (9 °C) to flow into the mixing bowl, thus forming a vortex. The powdered base ingredient (Table 2) was simultaneously fed from above and dispersed into the liquid stream. The base flowed down to the second mixing bowl aided by an impeller, where powders (extra protein and/or dietary fiber) were dispersed into the formed vortex and were pushed into the third mixing bowl by a second impeller. At the third mixing bowl, thickener and/or berry powder were dispersed into the formed vortex, mixed with a third impeller, and flowed-out by gravity into the serving cup (Table 1) followed by optional garnish sprinkle. After removal of the serving cup, a cold-water rinse flushed the line and was collected into a wastewater tank. In addition, hot water (99 °C) rinsed from a built-in boiler was programmed to take place every six hours or when idle for food safety.

Snack ordering in the HSM was operated via a user interface (UI) linked to a cloud service. The UI was accessible via web browser, which retrieved the specific recipe, ingredient, and nutritional information corresponding to user choice from the Miils.com web server. Communication between Miils.com and the HSM was executed via a Mosquitto MQTT broker server running on the Azure cloud service. Proprietary interface software was developed and installed on the HSM’s industrial PC, which subscribed to messages from Miils.com via the Mosquitto MQTT server. The interface software exchanged user choice commands with the PLC via an Automation Device Specification (ADS) interface, allowing the specific processing sequence to be carried out to prepare the snack. In addition, the interface software published status messages to Miils.com via Mosquitto MQTT. This solution was chosen to enable remote ordering and ease of expansion, as communication with several machines was possible via the same Mosquitto MQTT server. The UI choice menu is illustrated in Figure 3 and in Appendix A (Appendix A). The starting page of the developed user interface includes two options for the user; ’suggestions’ or ’create your own’ snack route. Suggestions included six pre-prepared options. In the ’create your own’ option (Figure 3), the user was able to customize the snack by choosing the consistency (spoonable or drinkable), the base (dairy or oat), and the flavor of the base (cocoa, vanilla, or berries). If an oat-based snack was chosen, the user was able to include additional protein (pea) and/or dietary fiber (digestion-resistant maltodextrin). Similarly, if dairy-based was chosen, additional protein (whey) and/or dietary fiber (digestion-resistant maltodextrin) could be included in the portion. Portion sizes available included small (1 dL), medium (2 dL), and large (3 dL). Finally, the user was able to choose to add a garnish sprinkled on top between ’coconut & chia’ and ’cranberry-strawberry granola’ options. The user was presented with the nutritional content and ingredient details of the customized product synchronously when the user made the choices with the UI before placing the order.

### 2.2. Evaluation of Snacks and HSM by Consumer Tests

#### 2.2.1. Consumer Liking of the Snack Products by Qualitative Testing

The consumer liking of selected cocoa-flavored spoonable samples from the descriptive sensory profiling sample set were studied at a shopping center in Espoo, Finland. The factors of interest were (1) the effect of protein and fiber addition, and (2) the effect of product base (oat versus dairy base). The inclusion criteria for the consumers were the regular use of snack products such as cereal bars and smoothies (at least once a month) and at least 15 years of age. Oat- and dairy-based samples were presented on separate days, forming a between-subjects design for the base mixtures and within-subjects design for the protein and fiber addition. The studied liking modalities of odor, appearance, texture, taste, and overall liking were measured with the 9-point hedonic scale. Samples were prepared by making single portions with the HSM, which were then combined, divided into 20 g samples in closed plastic cups, and refrigerated for up to one hour before evaluation. The samples were presented as “oat/dairy-based cocoa-flavored snack product prototypes.” At the beginning of the consumer testing, the consumers were informed that the tasting was related to research on a snack machine that could prepare snacks made to order in about 30 s. No information was given regarding the user interface, nutritional information, customization options, nor the appearance of the finished product. The sample order was randomized, and three-digit codes were used for the samples. Additional questions were related to sample usage scenarios and preferred portion size. The consumers were also asked to give feedback and improvement ideas on the snack prototype. Altogether, 55 consumers evaluated the oat-based samples (15–65 years, median age 31, 47% males) and 55 consumers the dairy-based samples (16–68 years, median age 28, 49% males). The results were analyzed using paired and independent sample t-tests for the liking data and *χ*^2^ tests for the willingness to buy, sample preference, and product size questions using SPSS version 26.

#### 2.2.2. Consumer Experience of HSM by Qualitative Testing

Consumer experience of the HSM prototype was studied in a real-life environment in the lobby of an office building. The HSM was placed next to a coffee machine and a vending machine, i.e., in an area where people are expected to purchase snacks and refreshments. The lobby area is open to both visitors and employees, who were all invited to use and evaluate the HSM. The office building in question was selected because it did not have its own cafeteria with the subsequent assumption that people working in the building typically needed to go out to have lunch. The HSM testing by consumers was performed during two successive working days between 10 a.m. and 4 p.m. The consumers were recruited without any specific recruitment criteria among the people walking through the lobby or using the existing vending machines, and they were asked to test a machine producing healthy, tailored snacks. During the two days of testing, 33 consumers (age distribution: 18–29 years, 3; 30–39 years, 10; 40–49 years, 9; 50–59 years, 8; over 60 years, 3; and 39% of the participants were male) completed the test with the HSM prototype. The usability study consisted of ordering the snack from the HSM by using the UI of the machine, testing and evaluating the snack, and answering an anonymous online survey (Appendix A (Appendix A)) with a laptop provided at the testing site. With the help of the survey, consumers evaluated, for example, the use of the HSM; the UI; and the look, smell, and taste of the snack. The time used to make the order on the UI was recorded, as well as any comments or actions during the process of ordering.

## 3. Results and Discussion

### 3.1. Snack and HSM Development

Drinkable and spoonable model snack products were developed by mixing powdered food ingredients with water in order to enable the working prototype of the HSM (Figure 2c) for instant production of healthy snack products at the point of purchase. The final prototype of the HSM consisted of individual ingredient containers from which powders are dispersed into water through a series of mixing bowls to enable on-demand preparation of several snack combinations (Figure 2b). Snack ordering took place through the UI, where the user was able to select a dairy- or oat-based snack and decide the flavor, consistency, protein, and/or dietary fiber supplementation and a garnish sprinkled on top (Figure 3). Upon ordering, the selected snack combination was communicated to the HSM via the cloud, which initiated the preparation of the snack by combining ingredients from different containers. The HSM hardware for powder reconstitution was designed to dose low-viscosity ingredients at the top levels, and the thickening ingredients dispersed at the final stage, where they dispersed into the liquid and developed viscosity fully in the serving cup. Thickening in spoonable recipes was achieved with a 3:1 mixture of heat-treated oat bran concentrate (enriched in beta-glucan) and CWS maize starch.

Supplementation with whey protein and fiber reduced the viscosity of spoonable samples significantly from 3.6 ± 0.2 to 0.6 ± 0.2 Pa s when analyzed 30 min after preparation (Table 1). The corresponding samples remained relatively stable afterward when evaluated at 60 min (3.6 ± 0.2 Pa s without supplementation, and 0.7 ± 0.2 Pa s with supplementation). In the presence of xanthan gum, cross-linked CWS waxy maize starch resulted in lower hydration rate and end viscosity compared to use of starch alone [15]. This effect was attributed to competitive hydration between gum and starch, restricting full swelling of the starch. It is likely that a similar mechanism occurred in the present study when nonviscous whey protein and dextrin were included. Lower viscosity is partly due to competitive hydration between whey protein, dextrin, starch, and beta-glucan and partly due to increased powder-to-water ratio limiting the total water availability (Table 1) and subsequent swelling and viscosity development of the starch. According to Calton et al. [12], apparent viscosities between 0.1 and 1 Pa s constitute soft and between 1 and 4 Pa s a semi-solid consistency. Therefore, protein and dietary fiber supplementation of the dairy-based snack alters the consistency from semi-solid to soft, although both can be considered spoonable. Similarly, addition of pea protein and dietary fiber reduced the viscosity significantly from 1.9 ± 0.2 to 0.4 ± 0.2 Pa s 30 min after preparation compared to in the oat-based cocoa spoonable sample. However, the viscosity in supplemented samples increased slightly to 2.3 ± 0.2 Pa s 60 min after preparation, indicating viscosity development as a function of storage time, whereas the viscosity of nonsupplemented samples remained stable (0.5 ± 0.2 Pa s). Viscosity of oat-based samples were also lower when compared to dairy-based samples. The differences in the viscosity development are probably due to a combination of different starches present (extruded oat, waxy maize) and their swelling rate, as well as the hydration and solubilization of beta-glucan, which can also aggregate and gel over time [16].

In general, oat-based recipes required more adjustment of the recipe to function in the HSM. The pea protein isolate (PPI) used increased the viscosity of the solution alone and synergistically with other ingredients, such as the oat base and thickener mix. For this reason, PPI was dosed at a moderately low level at 2.2 g (Table 1) along with a lower level of thickener in the supplemented recipe to avoid premature viscosity development and subsequent blockage of the mixing line. The lower viscosity in the supplemented recipe is due to less thickener present, as well as lower water availability compared to the nonsupplemented recipe.

### 3.2. Sensory Profiling and Consumer Liking of Snacks

The sensory profiles of both drinkable and spoonable samples showed that the oat-based samples differed from the dairy-based in multiple attributes, such as sweetness, bitterness, sedimentation tendency, and cocoa flavor (Table 3, and Appendix A (Appendix A)). Dairy-based samples were sweeter and more intensely cocoa-flavored. This is likely due to small differences between the recipes: While identical amounts of cocoa were used in the snack bases, the dairy snacks contain more of the snack base (Table 1), and thus, relatively more cocoa. Additionally, the skim milk powder of the dairy base had more sugars than the oat base.

Overall, the dairy-based samples were quite similar to each other despite the differences in protein and dietary fiber. The oat-based samples differed more between one another, for example, those with added protein and fiber were more bitter but less intense in grain-like flavor. The consistency of spoonable samples could be controlled well with protein and fiber additions. However, these additions resulted in dried pea odor and flavor for the oat-based samples, and other flavor (commented as milk powder-like) in dairy-based samples. The oat-based drinkable samples had a higher sedimentation tendency than the dairy-based samples. Investigations into increasing solubility or adding stabilizers to improve the stability of the oat-based products were not conducted, but the focus was more on finding a prototype recipe where the sample was fluid enough to pass through the mixing bowls in the HSM hardware, as described in Calton et al. (2019). Further, as the target was to produce snacks for instant use, high colloidal stability can be considered of less importance than in stored goods.

For the oat-based drinkable samples, Chakraborty et al. [17] published a systematic sensory study on the influence of varying contents of oat fiber in the mouthfeel and texture properties in model beverages. In their study, increased contents of both oat bran and milled oats resulted in increased thickness, mouth coating, sliminess, and dryness, which are in line with the results of the oat-based samples in the present study. Perceived sweetness did not increase in the 0–1.5% addition range of whey protein isolate in barley beta-glucan model beverages, whereas sourness, whey flavor intensity, and viscosity increased as a function of whey protein isolate content [18]. They investigated the odor-contributing volatile compounds in pea protein beverages. Nonheat-treated pea protein beverages had a significant beany odor, as well as pasta and potato odors [19].

According to the consumer liking test (Figure 3), the sample odor was the most liked part in the sample snacks (values 5.9 to 6.2, like slightly), while taste was least liked (4.1 to 4.9, ranging from dislike slightly to indifferent). However, there were no statistically significant differences in any liking modality between the protein and fiber-enriched sample and the nonenriched sample in neither oat-based (*t*(54) = 0.344 to 1.97, *p* > 0.05) nor dairy-based (*t*(54) = −1.91 to 0.10, *p* > 0.05) samples. This indicates that the recipe development was successful when targeting similar product consistencies regardless of the customization choice. Temelli et al. [18] similarly reported that the addition of whey protein isolate up to 1.5% (*w*/*w*) had no influence on the overall acceptability of the orange-flavored model beverages. The dairy samples received higher ratings than the oat-based samples (Figure 4) in appearance for the nonenriched samples (*t*(108) = 2.04, *p* = 0.044) and in taste for the protein and fiber-enriched samples (*t*(108) = 2.39, *p* = 0.018). While comparative hedonic studies on similar snacks are scarce, the result is in line with previous consumer studies that have compared dairy milks and nondairy milk substitutes; dairy versions are typically better liked [20,21]. Palacios et al. [21] reported that one segment in their consumer trial preferred the sweeter taste of the dairy milk compared to soy milk. However, the liking of oat-based milk substitutes in relation to dairy milk has been reported to increase after repeated exposure [22]. The lower liking in taste of the protein- and fiber-enriched oat-based sample can also be due to the beany flavor that is typically an undesired attribute in dairy alternatives [23]. There were no statistically significant differences in other liking modalities between oat- and dairy-based samples.

There were no differences in the willingness to buy between oat- and dairy-based samples (*χ*^2^ (6, *N* = 110) = 2.83, *p* = 0.83; Appendix A (Appendix A)). The preferred portion sizes were approximately even between 1 and 2 dL (Appendix A (Appendix A)). The most common usage scenarios were “between work/school and hobbies” (selected by 66% of respondents as one of three main usages, average rank 1.6) and “in transit from one place to another” (46% of respondents, average rank 1.7). Participants indicated that samples would require more flavor and sweetness in general, and their expectation in terms of odor and appearance from a chocolate flavoring was not met. It is likely that the combination of an unfamiliar sample and mild flavor in comparison to familiar and relatively clear chocolate odor and color resulted in a hedonic contrast [24] that decreased the taste scores. Several participants also perceived the samples as healthy, where the inclusion of nutritional and health benefits in the product information would have increased the ratings of the samples. Health benefits are often cited and suggested as a driver to increase use and liking of nondairy milk substitutes [20,25].

Based on the sensory profiling of snacks and consumer liking results, the recipes of the snacks were updated for the HSM prototype testing. The sugar and cocoa content were increased to boost the flavor and color (for cocoa) of the products (Table 2), and the pea protein preparation was also changed for milder pea taste.

### 3.3. Consumer Perceptions on HSM via Qualitative Real-Life Experimenting

In HSM prototype testing, consumers spent, on average, a minute tailoring their own snack with the machine. Five participants explored the options carefully, trying different combinations before settling on their final choice. In the end, the customized snacks were quite equally divided: 17 participants chose a drinkable product, and 16 a spoonable one (Table 4). The dairy and oat bases were equally popular with 16 choices for both (one user did not remember their choice). All the four combinations of the previous choices were almost equally popular. The biggest difference was in the customization of taste: 23 participants chose the berry flavor with only 7 opting for the cocoa and 2 for the vanilla. Apart from two people who did not remember their tailoring choices, all participants made at least one addition (fiber, protein, and/or garnish) to their snack. Similarly to the consumer liking study, the development suggestions were mostly about the taste. Participants also stated the wish for more tailoring options. The freeze-dried berries were a valued option, as they were evaluated to provide an element of freshness, natural flavor, and sweetness. According to Sowers et al. (2019), taste and healthfulness are the most important factors affecting consumers’ likelihood to purchase the product. In this study, the lowest scores were for the taste, smell, and the product’s ability to alleviate thirst or hunger. The lowest scores (below 3 on a 5-point Likert scale) for the taste came proportionally more often from participants who had chosen either the cocoa or vanilla base (3 for berry, 3 for cocoa, and 1 for vanilla). In the liking test, the participants wanted the portion size to be 1 or 2 dL. In the real-life test, the portion size was about 1 dL, so the relatively small size can explain why participants did not consider the snacks to replace an actual meal and alleviate hunger or thirst.

According to the survey results (Figure 5), the UI of the HSM was perceived to be clear, easy to use, and the ordering process was quick. The user was presented with the nutritional content and ingredient details of the customized product before placing the order. According to the participants, there was enough nutritional information available and the product seemed healthy for them. The availability of nutritional information has usually been mentioned by consumers as an important factor affecting their purchasing decision [7,26,27]. Still, previous studies found out that there is a big number of users who do not follow the nutritional information at all [7,28], and hunger and convenience drive the purchases made at vending machines [29,30]. In addition to nutritional information, allergens and information of the origin of the ingredients were previously reported to be valuable for the consumers [7].

When specifically asked, the participants of the present study were generally interested in having the possibility to use a similar machine in the future. The vending machines currently in their premises offer only pre-packaged snacks or candy, and this was seen as a more appealing and healthier option. In the previous studies as well, consumers have pointed out the importance that healthy snacks are available [4,5,7]. The decreased availability of high-calorie snacks has the biggest effect on consumption of low-calorie snacks over calorie labeling, increasing accessibility of low-calorie choices, and increasing prices of high-calorie choices [31]. However, Pharis et al. (2018) and Pechey et al. (2019) found that the availability of healthy snacks increased their consumption but resulted in revenue losses due to decreased sales volume [32,33].

The consumer test in the present study was conducted in an office building lobby, and that was also the location where the participants would have placed the machine. Other possible locations that were mentioned were the cafeteria and break rooms in the same office building. The office is part of the food environment for adults, and therefore, there is potential to improve dietary intake at the population level [34]. However, in our previous study, users defined the occasions for using the HSM when people are on the move, traveling, in a hurry, on their way to work or school, replacing lunch with a healthy snack on a busy day at work, or snacking in the afternoon on the breaks during and between work and hobbies [7]. In the present study, only three participants suggested locations outside the office: Public transport stations, gyms, and sports arenas. Therefore, it seems that the location of the study limited the ideation by participants, and thus, in the future, it would be valuable to study how much the study location affects consumer suggestions for new vending machine locations. In addition, vending machine density is usually high in school environments [35], which are also an interesting case for the HSM in the future. Vending machines in school environments have been studied previously, and the interest toward healthy snacks has also been pointed out in these studies [4,31,36]. In the present study, the preferred use situation in offices would be in the afternoon, either when the workday is long or when you are heading out to the gym after work. Even though the participants did not necessarily agree with the statement that the product took away hunger (Figure 4), they still felt the healthy snack could replace lunch on days when there was no time for a proper break. When asked about their preferred portion size (1/2/3 dL), six participants chose the 3 dL portion, one gave no answer, and the rest chose the 2 dL portion. This would be substantially bigger than the sample size used in the test and, on average, larger than the portion size preferred by users in the liking test. The size chosen might be influenced by the previous question (Question 12 in Appendix A), where participants were specifically asked how much they would pay for a 2 dL portion. However, the HSM is developed to enable the customization of snack products and portion size based on the needs and requirements of the consumer. The optimal portion size is affected, e.g., use situation, need of the energy, and product you choose. The nutritional content of the products varied between 48.8 and 95.0 kcal/100 g (Table 1).

For the willingness to pay, participants stated that a good price for the product would be about equal to what a store-bought snack (quark, yoghurt, or bottled smoothie) would cost. The price of healthier products has been discussed in previous studies as well. Price of the product affects purchasing behavior and can also be a potential barrier to buy healthy products [30,36]. Naturally, a lower price would increase consumers’ interest to purchase the healthier product [4,26,37]. Still, it is good to keep in mind that price reduction is not sustainable in the long term [38]. Consumers usually expect that healthier products are more expensive, and their willingness to pay more depends on the value of health they expect to receive from the product [26]. Thus, personal preferences and attitudes related to health and value consumers expect to have by purchasing healthy products affect the price they are ready to pay [26].

## 4. Conclusions

The study showed that the consumer perception toward customized and healthy snacks produced at the site of purchase was generally positive. We have previously explored the healthy snacking concept by using mock-ups [7] and evaluated the technical requirements [12], but in the present work, we focused on a real-life environment by prototype testing where consumers were able to order and test real snack products. We demonstrated that customized healthy drinkable and spoonable snack production by protein and/or fiber supplementation via various powder-based ingredients is possible with the prototype machine. We were able to prove that the previous findings based on the mock-up studies were in line with the functioning snack machine, and consumers were generally attracted by the HSM concept, although the sensory quality of products was not fully optimized. In future studies, product formulation with regard to the sensory quality with all available flavors is an important development target. However, when moving toward commercialization, ensuring food safety by feasible maintenance of the HSM is a critical technical requirement that should be addressed in future studies. The implementation of various clean-in-place systems and verification of their effectiveness in daily operation could be considered. The HSM prototype is a unique solution that is valuable for the study, focusing on the development concept of the customizable food products. Although the powder-based process benefits from long shelf life in ambient storage conditions, there are several options to prepare the snacks at the consumer interface in the future. In general, the results pave the way for viable solutions that can promote healthier snacking, and thus, improve public health.

## Figures and Tables

**Figure 1 foods-09-01454-f001:**
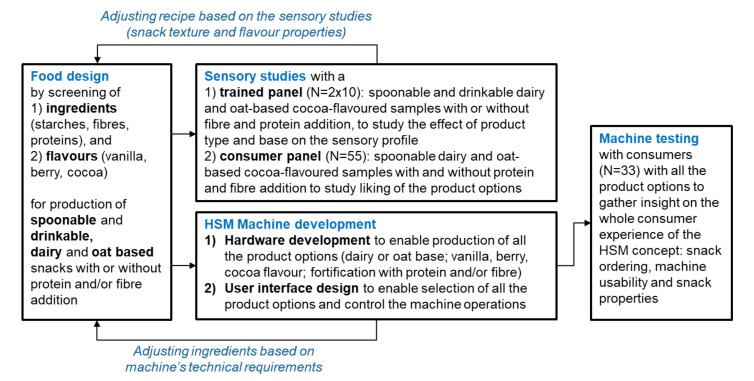
Scheme of the experimental design.

**Figure 2 foods-09-01454-f002:**
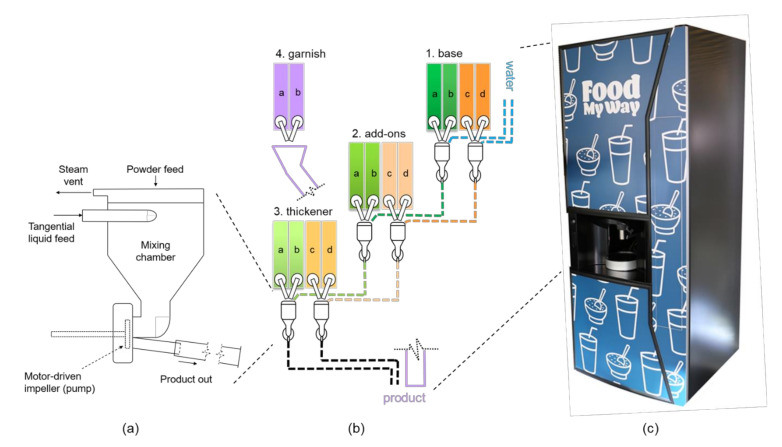
(**a**) Operating principle of motorized mixing bowl, (**b**) internal view of the Healthy Snack Machine (HSM) hardware showing operating principle (“base” signifies the flavor options: 1a = dairy base vanilla, 1b = dairy base cocoa, 1c = oat base vanilla, 1d = oat base cocoa; “add-ons” signify dietary fiber and protein options: 2a = pea protein isolate, 2b and 2c = digestion-resistant maltodextrin, 2d = whey protein concentrate; “thickener” signifies cold-swelling texturants and berry powder: 3a and 3d = berry powder, 3b and 3c = oat bran concentrate-waxy maize starch mixture (3:1); “garnish” signifies toppings sprinkled into serving cup: 4a = coconut-chia mixture, 4b = cranberry-strawberry granola), (**c**) photo of the functioning HSM prototype.

**Figure 3 foods-09-01454-f003:**
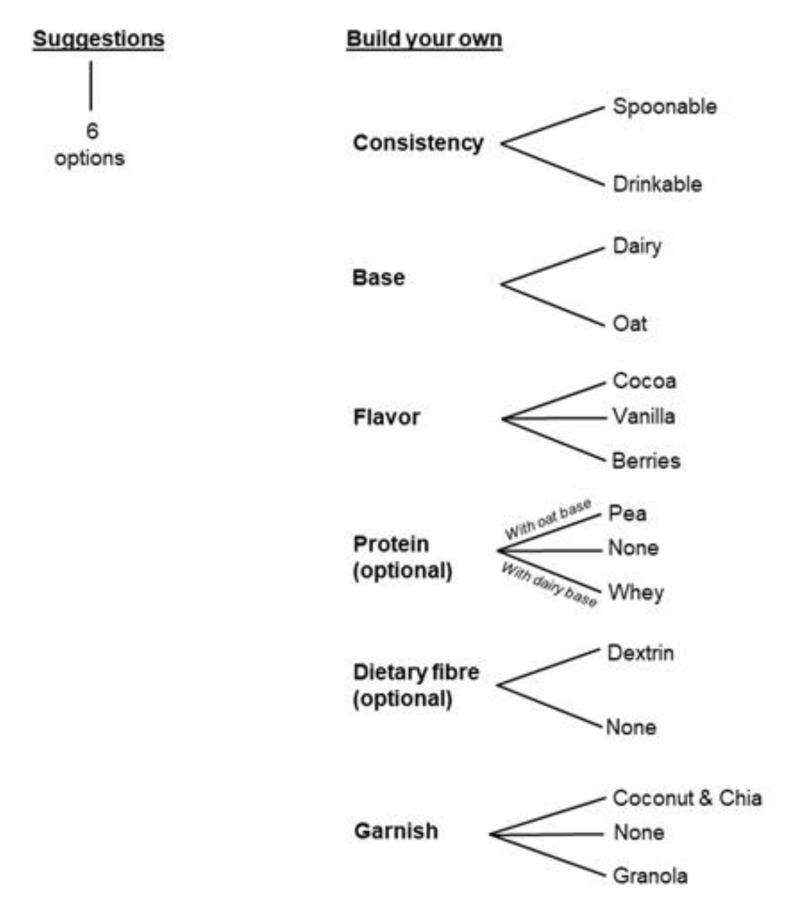
Illustration of the snack selection diagram in the HSM user interface with the six pre-prepared suggestions or ’create your own’ choice tree.

**Figure 4 foods-09-01454-f004:**
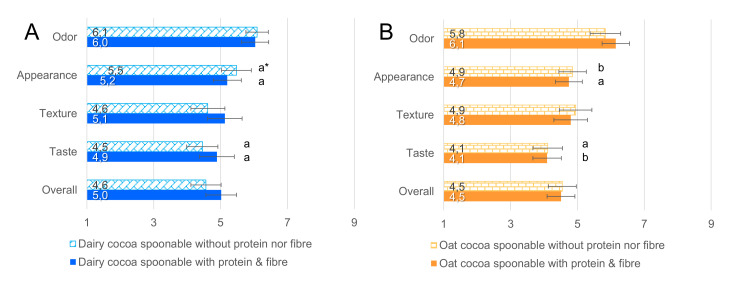
Bar charts of the average liking of dairy-based (**A**) and oat-based (**B**) cocoa-flavored spoonable samples. The charts are based on 55 responses. The 9-point hedonic scale was used for the evaluation of sample liking. Error bars are the 95% confidence intervals. * Sample types marked with different letters a,b have a statistically significant difference between the likings of respective dairy- and oat-based samples (*p* < 0.05, independent samples *t*-test).

**Figure 5 foods-09-01454-f005:**
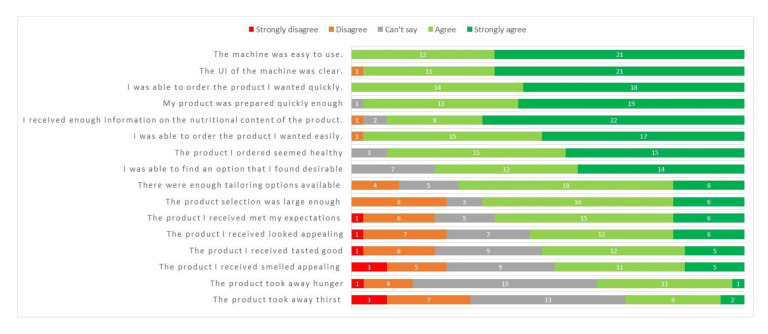
Questionnaire answers by consumers (*N* = 33) in the qualitative consumer survey of Healthy Snack Machine (HSM).

**Table 1 foods-09-01454-t001:** Recipes, nutritional information, and viscosity of cocoa spoonable and drinkable snacks used in sensory profiling and consumer trials. Water content in all samples is 100 mL. Mean values ± standard error followed by a different letter indicate statistically significant differences between samples at *p* < 0.05 (Tukey’s honest significant difference). Those that qualify for ’source of protein’ and ’source of fiber’ nutrition claims are underlined, and those ’high in protein’ or ’high in fiber’ are in bold ^1^. DF = dietary fiber, E = energy, na = not analyzed.

Product	Specifications	Base (g)	Added Protein (g), ^2^	Added DF (g), ^3^	Added Thickener (g), ^4^	E from Protein (%)	DF (g/100g)	kcal/100g	Apparent Viscosity (Pa s)	
									30 min	60 min
Spoonable	Dairy	18.6			11.1	**31.1**	**2.5**	77.1	3.6 ± 0.2 ^a^	3.6 ± 0.2 ^a^
	Dairy, protein, DF	18.6	6.8	4.2	11.1	**38.5**	**4.9**	95.0	0.6 ± 0.2 ^b^	0.7 ± 0.2 ^b^
	Oat	15.4			9.2	12.4	**2.0**	65.0	1.9 ± 0.2 ^c^	2.3 ± 0.2 ^c^
	Oat, protein, DF	15.4	2.2	4.2	6.6	18.1	**4.7**	74.6	0.4 ± 0.2 ^d^	0.5 ± 0.2 ^d^
Drinkable	Dairy	18.6				**40.3**	0.5	51.6	na	na
	Dairy, protein, DF	18.6	6.8	4.2		**47.5**	**3.2**	73.2	na	na
	Oat	15.4				10.7	0.7	48.8	na	na
	Oat, protein, DF	15.4	4.3	4.2		18.6	**3.6**	59.7	na	na

^1^ A claim that a food is a source of protein can be made when over 12% of the energy value is provided by protein. A claim that a food is high in protein can be made when over 20% of the energy value is provided by protein. The source of fiber claim can be made when the product contains a minimum of 3 g fiber per 100 g or 1.5 g fiber per 100 kcal. The high-in-fiber claim can be made when the product contains a minimum of 6 g fiber per 100 g or 3 g fiber per 100 kcal [14]; ^2^ whey protein concentrate for dairy-based and pea protein isolate for oat-based; ^3^ digestion-resistant maltodextrin; ^4^ a 3:1 mixture of extruded oat bran concentrate (15% beta-glucan) and cold water swelling waxy maize starch.

**Table 2 foods-09-01454-t002:** Ingredient composition (%) of oat and dairy snack bases before (A) and after (B) revision. OF = Oat flour; SMP = Skim-milk powder; DME = dry malt extract. Berry flavor was not formulated as a separate base but was produced by adding berry powder to vanilla-based snacks.

Oat (A)	Vanilla	OF (55.0%)	DME (44.0%)	Vanilla aroma (1.0%)		
	Cocoa	OF (47.4%)	DME (44.0%)	Cocoa (8.0%)	Salt (0.6%)	
Dairy (A)	Vanilla	SMP (70.0%)	DME (29.0%)	Vanilla aroma (1.0%)		
	Cocoa	SMP (61.4%)	DME (30.0%)	Cocoa (8.0%)	Salt (0.6%)	
Oat (B)	Vanilla rev.	OF (55.0%)	DME (40.0%)	Vanilla sugar (5.0%)		
	Cocoa rev.	OF (42.0%)	DME (35.4%)	Cocoa (20.0%)	Vanilla sugar (2.0%)	Salt (0.6%)
Dairy (B)	Vanilla rev.	SMP (71.0%)	DME (24.0%)	Vanilla sugar (5.0%)		
	Cocoa rev.	SMP (60.0%)	DME (17.4%)	Cocoa (20.0%)	Vanilla sugar (2.0%)	Salt (0.6%)

**Table 3 foods-09-01454-t003:** Two-way mixed-model ANOVA results (averages and standard deviations) for the drinkable and spoonable samples.

Sample	ANOVA *p*	Partial *η*^2^	Dairy with Protein and Fiber			Oat with Protein and Fiber			Dairy without Protein nor Fiber			Oat without Protein nor Fiber		
**Drinkable samples**														
Cocoa odor	<0.001	0.53	4.2	(1.4)	a	2.6	(1.3)	c	3.9	(1.3)	ab	3.1	(1.3)	bc
Sweet odor	<0.001	0.64	6.7	(1.4)	a	3.2	(1.8)	b	5.8	(1.6)	a	4.0	(1.8)	b
Grain-like odor	0.003	0.40	1.5	(1.5)	c	2.7	(1.7)	ab	2.0	(1.5)	bc	3.3	(1.5)	a
Dried pea odor	<0.001	0.67	1.3	(1.6)	b	5.1	(2.1)	a	1.5	(1.4)	b	1.8	(1.7)	b
Sedimentation tendency	<0.001	0.83	1.9	(1.4)	c	5.5	(1.8)	b	2.5	(1.7)	c	7.4	(1.4)	a
Sweetness	<0.001	0.88	5.0	(1.4)	a	2.3	(1.2)	b	5.3	(1.6)	a	2.0	(1.3)	b
Bitterness	<0.001	0.55	2.1	(1.8)	c	5.5	(2.0)	a	2.6	(1.7)	bc	3.4	(1.6)	b
Cocoa flavor	<0.001	0.76	4.8	(1.0)	a	1.7	(0.9)	c	5.0	(1.0)	a	3.1	(1.6)	b
Grain-like flavor	0.003	0.40	2.3	(1.8)	b	2.3	(1.8)	b	2.3	(1.8)	b	4.1	(1.9)	a
Dried pea flavor	<0.001	0.87	1.7	(1.6)	b	6.9	(1.7)	a	1.4	(1.5)	b	1.9	(1.9)	b
Richness	<0.001	0.51	4.3	(2.2)	a	4.4	(2.0)	a	4.3	(2.1)	a	1.8	(1.6)	b
Astringency	0.007	0.36	2.0	(1.0)	b	3.6	(1.9)	a	1.8	(1.5)	b	2.4	(1.4)	b
Other flavor intensity	<0.001	0.63	4.5	(2.4)	a	1.3	(2.7)	b	3.8	(2.5)	a	0.9	(1.6)	b
**Spoonable samples**														
Cocoa odor	<0.001	0.66	4.0	(1.4)	a	2.3	(1.4)	c	4.1	(1.0)	a	3.1	(1.2)	b
Grain-like odor	0.08	0.22	2.5	(1.7)		2.7	(1.8)		2.6	(1.7)		3.6	(1.5)	
Dried pea odor	<0.001	0.62	1.2	(1.4)	b	4.3	(2.3)	a	1.3	(1.5)	b	1.6	(1.8)	b
Stretchability	0.008	0.35	3.7	(1.8)	b	5.3	(0.9)	a	4.0	(1.5)	b	3.5	(1.8)	b
Consistency	<0.001	0.75	1.8	(0.9)	c	3.1	(0.7)	b	4.4	(1.1)	a	4.7	(1.5)	a
Graininess	0.025	0.29	5.2	(1.8)	a	3.8	(1.3)	b	4.2	(1.4)	b	4.1	(1.3)	b
Sliminess	<0.001	0.68	2.8	(2.5)	b	5.5	(2.1)	a	5.7	(2.1)	a	6.0	(2.0)	a
Sweetness	<0.001	0.65	5.1	(1.3)	a	2.5	(1.4)	b	4.5	(1.3)	a	2.5	(1.9)	b
Bitterness	<0.001	0.54	2.1	(1.4)	bc	4.4	(2.0)	a	1.9	(1.5)	c	2.6	(1.9)	b
Cocoa flavor	<0.001	0.73	4.9	(1.1)	a	2.0	(1.2)	c	4.6	(0.9)	a	3.1	(1.4)	b
Grain-like flavor	0.027	0.28	3.5	(1.7)	bc	3.0	(1.7)	c	4.1	(1.8)	ab	4.7	(1.6)	a
Dried pea flavor	<0.001	0.69	1.3	(1.7)	b	5.5	(2.1)	a	1.6	(1.8)	b	1.4	(1.3)	b
Richness	0.020	0.30	4.7	(2.2)	b	5.0	(1.6)	b	6.7	(0.8)	a	5.2	(2.1)	b

The samples marked with different letters a–c mark statistically significant differences between samples for each attribute in Tukey’s post-hoc tests; group a has the largest intensity.

**Table 4 foods-09-01454-t004:** Consumer choices (*N* = 33) in the user interface during the qualitative study. “No info” indicates the five people who chose the quick choice, and thus, were not able to choose garnishes. Further, one person chose the quick choice and did not answer any questions on the tailoring.

Choice Category in the Machine Operating Principle (Figure 3)	Operation Presented to Consumer	Number of Consumers Choosing the Option
Viscosity	Drinkable	17
	Spoonable	15
Base	Oat	16
	Dairy	16
Flavour	Cocoa	7
	Vanilla	2
	Berries	23
Extra protein	Yes	11
	No	16
	No info	5
Extra fibre	Yes	12
	No	15
	No info	5
Garnish	Coconut & chia	6
	Granola	11
	None	10
	No info	5

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
