# Peer review of "Development and Consumer Perception of a Snack Machine Producing Customized Spoonable and Drinkable Products Enriched in Dietary Fiber and Protein"

_foods, 2020, doi:10.3390/foods9101454_

Round 1
Reviewer 1 Report
This manuscript entitled "Development and consumer perception of a snack machine producing customized spoonable and drinkable products enriched in dietary fiber and protein" is precisely analyzed the consumer perceptions for customized snacks produced with HSM. Most of the data are well-mentioned. However, the following information is difficult to understand by writing only. The author should explain it by table or figure.
Line 15 & 253: The viscosity of samples is not shown in Tables or Figures. It is difficult to compare various data values in the manuscript only. Please show it by table or figure.
Author Response
Dear Reviewer,
Thank you very much for your valuable feedback related to our manuscript "Development and consumer perception of a snack machine producing customized spoonable and drinkable products enriched in dietary fiber and protein". We have now submitted the latest version of the manuscript for revision.
Here, you can find our response to your feedback:
Point 1: Line 15 & 253: The viscosity of samples is not shown in Tables or Figures. It is difficult to compare various data values in the manuscript only. Please show it by table or figure.
Response 1: To improve the clarity, we have now added the data in Table 1 and updated the table caption also. In addition, ANOVA and Tukey’s HSD described on lines 137-138.
In addition to these updates, the feedback from other reviewers has been taken into consideration.
On behalf of co-authors,
Yours sincerely,
Kaisa Vehmas
Reviewer 2 Report
The strength of the paper is the innovation, the authors did an impressive work building the prototype of the machine.
I suggest for the paper to be separated into two separate submissions: 1) the prototype and interface and 2) the sensory evaluation of the products
Here are some comments and suggestions that will improve the quality of the paper:
Abstract: The abstract can be improved by explaining clearly the experimental design as it is not very clear what was the experiment about after reading this section
Introduction:
Line 32: the average of snacking in which region of the world?
Line 89: clarify the meaning of "1856 from Condetta"
Table 1: This table is not very clear as it doesn't take into consideration the vanilla, chocolate, and berries and the sprinkles added. You mentioned in the footnote that whey protein concentrate and pea protein isolate for dairy. Maybe you should explain the experimental design using a figure first and then go into more details so when the readers go over the paper they will fully understand the methods and results.
Table 2. You didn't include berry flavor and yet you mention it in the text (line 107)
For the descriptive sensory profiling justify why you only used cocoa-based products instead of vanilla and berry flavor as well.
Line 151: you mentioned "optional protein and/or dietary fiber". I understood from the text that every time you add protein you need to add fiber as most of the tables you have "protein and fiber" as a title. Please clarify this point in the different sections is its protein and fiber or protein and/or fiber (for example, line 166 you wrote fiber and protein)
I also have a concern about the cleaning procedure. You mentioned in line 161 that hot water rinses but only every six hours. From the food safety point of view, hot water should be used after each use to avoid the accumulation of biofilms and to clean the fat and proteins in the tubes as these compounds are not easily carried away with only room temperature water. This is something that should be studied in the future, making sure that the microbiology of the products is fine after each use.
Figure 1. The diagram is hard to follow, maybe you can improve it by using different colors.
Figure 2: the dietary fiber option under "add more" are blank
Results and discussion
Line 240: you mentioned that you mixed powdered food with water. Is this how you planned to prepare the snacks regularly? Or are you planning to use different liquids?
Line 254: statistical results missing
Figure 3: check statistical results for appearance and taste
I can observe from this figure that the liking was "neutral" as it was around 4.5. You might need to reconsider the formulation to improve the flavor and aroma
Line 356: you mentioned that you changed the pea protein to "enable lower viscosity and milder pea taste". At what point did you change it? would that affect the results? Did all the samples test contained the same pea flour?
Clarify what was your decision about the ideal final volume for the beverage
Author Response
Dear Reviewer,
Thank you very much for your valuable feedback related to our manuscript "Development and consumer perception of a snack machine producing customized spoonable and drinkable products enriched in dietary fiber and protein". We have now submitted the latest version of the manuscript for revision.
Here, you can find our response to your feedback:
Point 1: I suggest for the paper to be separated into two separate submissions: 1) the prototype and interface and 2) the sensory evaluation of the products
Response 1: We believe that for the main outputs and overall quality of the paper, i.e. to demonstrate the development and investigation of the new snack food development concept and technology, it is more valuable to include all the data in a single research paper. Indeed, by bringing in new food engineering technology with the sensory quality evaluated by sensory and consumer panels as well as the user experience of the new food manufacturing at the consumer interface by analyzing both UI and HSM machine, we can provide new insight to both food engineering and consumer research fields. To be able to complete this multi-disciplinary study approach, we decided to prioritize studying the effect of more fundamental differentiating factors, such as base type of snack (oat or dairy) and their customization with macronutrients in the sensory profiling and consumer testing. This limited the factors involved in sensory aspects of the article but we argue that this decision was for the benefit of the whole article entity (ref to the comments below). To clarify our approach, we have edited the abstract by including more information on our experimental design, and we have also made a schematic illustration of the study flow, see Figure 1 (ref also to the comment below).
Point 2: The abstract can be improved by explaining clearly the experimental design as it is not very clear what was the experiment about after reading this section
Response 2: We appreciate the comment and have attempted to improve the abstract in order to define the experimental design. Furthermore, to clarify the experimental design, a schematic illustration of the study flow (Figure 1) was added to assist understanding the methods and results used in the work. The scheme has been described on lines 76-80.
Point 3: Line 32: the average of snacking in which region of the world?
Response 3: These figures are from the United States. The information has been included in line 31.
Point 4: Line 89: clarify the meaning of "1856 from Condetta"
Response 4: This refers to the second type of cocoa powder along with D-11-CK from ADM International. This clarification has been added to lines 96-98.
Point 5: Table 1: This table is not very clear as it doesn't take into consideration the vanilla, chocolate, and berries and the sprinkles added. You mentioned in the footnote that whey protein concentrate and pea protein isolate for dairy. Maybe you should explain the experimental design using a figure first and then go into more details so when the readers go over the paper they will fully understand the methods and results.
Response 5: This Table 1 is to illustrate the nutritional quality of one of the recipes, i.e. the one with cocoa flavor. Adding other flavors did not change the nutritional content significantly, and therefore, the data is not shown for other flavors. This has now been clarified in the text (lines 89-90), and also the table caption was updated. In addition, the footnote in Table 1 (line 115) was updated to clarify that whey protein concentrate is an option for dairy-based only, and pea protein isolate for oat-based only. Furthermore, to clarify the experimental design, a schematic illustration of the study flow (Figure 1) was added to assist understanding the methods and results used in the work.
Point 6: Table 2. You didn't include berry flavor and yet you mention it in the text (line 107)
Response 6: This issue has been clarified in the caption of Table 2 and includes “Berry flavor was not formulated as a separate base but was produced by adding berry powder to vanilla-based snacks”. The berry flavor along with vanilla flavor were included in the HSM technological concept but were not included in the sensory profiling and consumer liking trials. However, they were featured in the user interface trial.
Point 7: For the descriptive sensory profiling justify why you only used cocoa-based products instead of vanilla and berry flavor as well.
Response 7: We decided to focus on the sensory profiling to detect differences in the snack product bases (dairy versus oat) as well as on spoonable and drinkable products when fortified with protein and/or fiber. The effect of flavoring was considered to be small on these two factors and choosing only one flavor allowed us to utilize complete block designs (i.e. all samples were evaluated in each session) for the drinkable and spoonable samples in the descriptive profiling. Cocoa also functioned similarly in the prototype instrument for all product formulations and did not affect the consistency, which made it a suitable choice for characterization.
Therefore, different flavors were not included in the sensory study to keep the experimental set up feasible and reasonable. A subset of the descriptive profiling samples (extreme configurations for spoonable samples) were selected for the consumer test. This has been described on line 141-150 and lines 221-223.
The different flavors were included in the consumer testing with the HSM machine testing to enable studying the personalization aspect as broadly as possible there, i.e. including different product bases (dairy, oat), types (spoonable, drinkable), fortification (fiber, protein) and flavor (berry, vanilla, cocoa). This has also been mentioned on line 476-477; all product categories will be studied in the future. To clarify the experimental design of the research, we have now added schematic presentation of the study parts (as mentioned already above) which also includes the sensory studies and their role in the new Figure 1.
Point 8: Line 151: you mentioned "optional protein and/or dietary fiber". I understood from the text that every time you add protein you need to add fiber as most of the tables you have "protein and fiber" as a title. Please clarify this point in the different sections is its protein and fiber or protein and/or fiber (for example, line 166 you wrote fiber and protein)
Response 8: The reviewer is right that in the sensory studies, the samples either had both extra protein and fiber or neither of them. This was due to our focus on comparing the extreme sample cases. However, with the machine it is possible to also select just either extra protein or extra fiber. These dimensions were not investigated in the sensory studies to keep the sample experimental design reasonable. We have clarified the relevant text section, e.g. check lines 141-151 and 205-209. Further, we have clarified the optional nature of fiber and protein choice in Figure 2.
Point 9: I also have a concern about the cleaning procedure. You mentioned in line 161 that hot water rinses but only every six hours. From the food safety point of view, hot water should be used after each use to avoid the accumulation of biofilms and to clean the fat and proteins in the tubes as these compounds are not easily carried away with only room temperature water. This is something that should be studied in the future, making sure that the microbiology of the products is fine after each use.
Response 9: This is indeed an important topic and we have considered it. We found that an immediate pre-rinsing with cold water (followed by hot-rinse) is also relevant since it minimizes heat-induced fouling by protein and carbohydrate components. In the current prototype, hot water rinses are actually programmed to take place every six hours or when the machine is in idle state (no orders being placed), as stated on lines 180-181. This means that hot water rinses occur more frequently than every six hours but is dependent on use frequency. Currently, without a clean-in-place system implemented (and automated verification of cleaning efficiency), disassembly and manual cleaning at regular intervals is still necessary. In any case, this is a very important topic for future studies, and thus, we have now included a sentence in the future perspective section about the CIP issue, on lines 478-480.
Point 10: Figure 1. The diagram is hard to follow, maybe you can improve it by using different colors.
Response 10: We have now improved the readability of the Figure 1 (now Figure 2) by using different colors.
Point 11: Figure 2: the dietary fiber option under "add more" are blank
Response 11: Figure 2 (now figure 3) is now corrected, the missing parts were included and modified a bit.
Point 12: Line 240: you mentioned that you mixed powdered food with water. Is this how you planned to prepare the snacks regularly? Or are you planning to use different liquids?
Response 12: There are several options to prepare the snacks at the consumer interface in the future. We selected the powder and water mix, since it was feasible for the equipment we had for the study. There are many benefits to using powders, such as long shelf life and storage in ambient temperature conditions. In the future, we envision including more sophisticated features, such as cold storage or liquids’ dispensing from aseptically packaged “bag-in-a-box" type solutions, which could offer better sensory properties also. However, such solutions are closely tied to establishment of supply chain and maintenance in a commercial operation, which we are currently working towards. This has been described on lines 482-484.
Point 13: Line 254: statistical results missing
Response 13: This viscosity data with statistics is now presented in Table 1.
Point 14: Figure 3: check statistical results for appearance and taste. I can observe from this figure that the liking was "neutral" as it was around 4.5. You might need to reconsider the formulation to improve the flavor and aroma
Response 14: The statistical results (letters a and b) refer to the comparison between oat and dairy samples (between-subjects test, independent t-test, values give in text row 345) and not the comparison within the protein&fiber vs non-enriched pair for dairy and oat samples. However, the reviewer is right that the marking is not making this point clear. We have clarified the markings in Figure 4 (previously Fig. 3) and added a footnote to better convey this comparison.
This is a correct observation from the reviewer. The taste was the lowest modality of liking and that likely drove the overall liking down as well. As we argue in rows 371-372 based on the open comments from the consumers, there seem to be two main reasons for the low taste ratings: 1) lack of full context in giving them the sample (without garnishes, just a small portion, and not telling that it has a porridge-like consistency) and 2) contrast between the first impression given by odor (intense cocoa odor) and the impression from taste (mild, porridge-like flavor). While flavor optimization was not the goal of this research, reconsidering the formulation is indeed an important point in future studies. We have added a related point to this in lines 475-477.
Point 15: Line 356: you mentioned that you changed the pea protein to "enable lower viscosity and milder pea taste". At what point did you change it? would that affect the results? Did all the samples test contained the same pea flour?
Response 15: The pea protein isolate was changed after sensory profiling, product characterization and consumer liking trials to a milder flavored alternative for the HSM prototype testing. This has been further clarified in the manuscript on line 122-123 and 381. We are of the opinion that this change allows better study of the overall concept in HSM prototype testing, since the negative flavor impact with pea protein fortification is lower and less distracting to the users. Furthermore, in HSM prototype testing all options were available for the users, and thus any possible effect from a different pea protein is limited to those choices that included pea protein fortification.
Point 16: Clarify what was your decision about the ideal final volume for the beverage
Response 16: HSM is developed to enable the customization of snack products and portion size based on the needs and requirements of the consumer. The optimal portion size is affected e.g. use situation, need of the energy and product you choose. The nutritional content of the products vary between 48.8 and 95.0 kcal/100g. This has been defined on lines 452-455.
In addition to these updates, the feedback from other reviewers has been taken into consideration.
On behalf of co-authors,
Yours sincerely,
Kaisa Vehmas
Reviewer 3 Report
The manuscript seems to me to be consistent, carefully crafted. Tables and Figures suitably complement the textual part of the manuscript and help to better understand the concept of the experiment.
Line 403: I miss the citation [33] (Pechey et al.) at the end of the sentence after the reference [32].
I have no further comments on the manuscript.
Author Response
Dear Reviewer,
Thank you very much for your valuable feedback related to our manuscript "Development and consumer perception of a snack machine producing customized spoonable and drinkable products enriched in dietary fiber and protein". We have now submitted the latest version of the manuscript for revision.
Here, you can find our response to your feedback:
Point 1: Line 403: I miss the citation [33] (Pechey et al.) at the end of the sentence after the reference [32].
I have no further comments on the manuscript.
Response 1: Corrected, the other reference number has been added on line 428.
In addition to these updates, the feedback from other reviewers has been taken into consideration.
On behalf of co-authors,
Yours sincerely,
Kaisa Vehmas
Round 2
Reviewer 2 Report
Dear authors,
The paper improved its quality with the changes you made since the last revision. Also, in your detailed response letter, you address all of my questions and these changes were reflected in the paper.